# The tricuspid valve also maladapts as shown in sheep with biventricular heart failure

William D Meador[1]*, Mrudang Mathur[2], Gabriella P Sugerman[1], Marcin Malinowski[3,4], Tomasz Jazwiec[3,5], Xinmei Wang[6], Carla MR Lacerda[6], Tomasz A Timek[3], Manuel K Rausch[1,2,7]*

[1]Department of Biomedical Engineering, The University of Texas at Austin, Austin, United States; [2]Department of Mechanical Engineering, The University of Texas at Austin, Austin, United States; [3]Division of Cardiothoracic Surgery, Spectrum Health, Grand Rapids, United States; [4]Department of Cardiac Surgery, Medical University of Silesia, School of Medicine in Katowice, Katowice, Poland; [5]Department of Cardiac, Vascular and Endovascular Surgery and Transplantology, Medical University of Silesia in Katowice, Silesian Centre for Heart Diseases, Zabrze, Poland; [6]Department of Chemical Engineering, Texas Tech University, Lubbock, United States; [7]Department of Aerospace Engineering and Engineering Mechanics, The University of Texas at Austin, Austin, United States

**Abstract** Over 1.6 million Americans suffer from significant tricuspid valve leakage. In most cases this leakage is designated as secondary. Thus, valve dysfunction is assumed to be due to valve-extrinsic factors. We challenge this paradigm and hypothesize that the tricuspid valve maladapts in those patients rendering the valve at least partially culpable for its dysfunction. As a first step in testing this hypothesis, we set out to demonstrate that the tricuspid valve maladapts in disease. To this end, we induced biventricular heart failure in sheep that developed tricuspid valve leakage. In the anterior leaflets of those animals, we investigated maladaptation on multiple scales. We demonstrated alterations on the protein and cell-level, leading to tissue growth, thickening, and stiffening. These data provide a new perspective on a poorly understood, yet highly prevalent disease. Our findings may motivate novel therapy options for many currently untreated patients with leaky tricuspid valves.

*For correspondence:
william.meador@utexas.edu (WDM);
manuel.rausch@utexas.edu (MKR)

## Introduction

The tricuspid valve regulates blood flow between the right atrium and right ventricle. Its function as a check-valve depends on the well-orchestrated interplay between the three leaflets and the valve's annular junction with the surrounding myocardium. During ventricular systole, the three leaflets coapt and seal the valve orifice, while, during diastole, they open to allow for blood to pass. Leaflet coaptation is further ensured through fibrous cords which connect the leaflets to the papillary muscles of the ventricular myocardium, akin to parachute cords that prevent leaflet prolapse into the atrium (*Silver et al., 1971*; *Smith et al., 2020*; *Meador et al., 2020a*). In tricuspid regurgitation (TR), leaflet coaptation is incomplete allowing for retrograde leakage through the valve (*Mangieri et al., 2017*).

Moderate to severe TR affects more than 1.6 million Americans and is an independent predictor of mortality (*Mangieri et al., 2017*; *Taramasso et al., 2012*; *Nath et al., 2004*). In 85% of severe cases, TR is considered functional or secondary to right ventricular remodeling and left-sided heart

disease (*Mangieri et al., 2017*; *Dreyfus et al., 2015*; *Rogers and Bolling, 2009*). Here, resultant tricuspid annular dilation and papillary muscle displacement are believed to circumferentially strain and tether the valve's leaflets, respectively, preventing proper leaflet coaptation (*Sun and O'Gara, 2017*; *Spinner et al., 2012*). In other words, it is believed that constraints from valve-extrinsic conditions render the valve dysfunctional, while the valve itself is considered structurally and mechanically intact and healthy (*Meador et al., 2018*; *Lee et al., 2019*). In fact, TR was historically not treated assuming that it would resolve following treatment of left-sided heart disease (*Braunwald et al., 1967*). However, this conservative approach to TR has since been reconsidered and treatment strategies are more aggressive, including invasive surgical repair or replacement. Despite decades of experience with these devices and techniques, repair failure rates are still as high as 10–30% (*Pfannmüller et al., 2012*). As such, TR remains a poorly understood disease with significant opportunity for therapeutic improvement.

Interestingly, on the left side of the heart, functional mitral regurgitation has been found to be not so functional after all (*Gillam, 2008*). Our group and others have shown that the mitral valve leaflets grow and remodel in functional mitral regurgitation (*Rausch et al., 2012*; *Dal-Bianco et al., 2009*; *Stephens et al., 2009*; *Grande-Allen et al., 2005a*). While leaflet growth, that is adaptation, in the mitral valve may be beneficial by increasing the coaptation area, (*Beaudoin et al., 2013*; *Dal-Bianco and Levine, 2015*) concomitant leaflet fibrosis, that is maladaptation, may stiffen the valve and thus impede proper kinematics and coaptation (*Grande-Allen et al., 2005b*). Thus, mitral leaflet (mal)adaptation, that is both adaptation and maladaptation, plays a diametrical role. Fortunately, Levine and co-workers have demonstrated that the maladaptive, fibrotic response of the leaflets may be pharmacologically reduced without reducing their adaptive ability to grow (*Bartko et al., 2017*). Thus, in the future, pharmacological management of mitral regurgitation targeting leaflet remodeling may support its surgical/interventional treatment and improve long-term outcomes (*Rausch, 2020*).

Similar research on the tricuspid valve is nearly absent, so much so that it has been dubbed the 'forgotten valve' (*Stephens and Borger, 2014*). Although a first study in humans demonstrated that leaflets of patients with pulmonary hypertension may increase in area, it remains unknown whether tricuspid valve leaflets similarly maladapt during functional TR (*Afilalo et al., 2015*). Whether they actively maladapt or not is ultimately important as it may, as is the case with the mitral valve, challenge the notion of TR being strictly functional. In turn, a better understanding of TR may inform improved surgical strategies or provide alternative treatment routes. Therefore, our objective was to fill this gap in knowledge and investigate whether the tricuspid valve, similarly to the mitral valve, (mal)adapts during TR.

## Results

### Animal model outcomes

We successfully isolated 17 tricuspid valves from control (CTL) sheep and 33 tricuspid valves from sheep that underwent 19 ± 6 days of rapid pacing to induce biventricular heart failure (TIC). We included only anterior tricuspid valve leaflets in our study because the anterior leaflet has the largest major cusp of the three leaflets which allowed us to conduct all (or most) of our analyses on one and the same tissue. Furthermore, because of its tethering to the remodeling right ventricular free wall we suspected that the anterior leaflet also maladapts the most of the three leaflets. Echocardiographic data and hemodynamic data of the animals revealed significant parameter changes in TIC subjects consistent with clinical biventricular dysfunction and functional tricuspid valve regurgitation (*Table 1*). Most notably, we observed that TIC subjects had significant biventricular ejection fraction reduction, biventricular remodeling, tricuspid valve annular dilation, and increased TR severity.

### Anterior leaflet area increases in TIC

Based on previous studies in the mitral and tricuspid valves, we hypothesized that anterior leaflets would increase in area in TIC animals. Toward testing this hypothesis, we measured the anterior leaflet's area, and major cusp height and width (*Figure 1a*). Once isolated, all anterior leaflets appeared anatomically typical (*Silver et al., 1971*). However, in our analysis we found a significant increase of approximately 130% in anterior leaflet area for TIC animals compared to CTL animals (p=0.002,

**Table 1.** Echocardiographic and hemodynamic data of animal model.

| Parameter | CTL | TIC (baseline) | TIC (terminal) |
|---|---|---|---|
| Echocardiographic data | | | |
| TR grade (i.e., severity) | 1.0 (0.0) | 0.0 (0.0) | 2.0 (1.0) * † |
| RV EF, % | 67.0 ± 8.6 | 58.8 (20.5) | 48.4 ± 12.9 * † |
| RV FAC, % | 52.1 ± 6.0 | 53.9 ± 8.0 | 37.4 ± 8.7 * † |
| RV IDd, cm | 2.5 ± 0.4 | 2.6 (0.6) | 2.9 ± 0.7 * |
| TV annulus dimension, cm | 2.5 (0.3) | 2.5 ± 0.4 | 3.2 ± 0.5 * † |
| MR grade (i.e., severity) | 0.0 (0.0) | 0.0 (0.0) | 2.0 (1.0) * † |
| LV EF, % | 55.8 ± 4.4 | 61.3 ± 6.2 | 30.0 (11.2) * † |
| LV IDd, cm | 4.2 ± 0.3 | 3.7 ± 0.6 | 4.6 ± 0.4 * † |
| Hemodynamic data | | | |
| HR, bpm | 108 (11) | - | 126 ± 23 * |
| RV $P_{Max}$, mmHg | 28.8 (11.4) | - | 44.0 ± 11.5 * |
| RV $P_{ES}$, mmHg | 22.3 ± 5.6 | - | 35.7 ± 11.0 * |
| RA $P_{Mean}$, mmHg | 10.5 (3.2) | - | 10.6 (5.6) |
| LV $P_{Max}$, mmHg | 96.1 ± 9.7 | - | 92.9 ± 13.8 |
| LV $P_{ES}$, mmHg | 63.1 ± 17.8 | - | 64.4 ± 13.6 |

Values are mean ± standard deviation or median (interquartile range).

CTL = control, EF = ejection fraction, ES = end systolic, FAC = fractional area change, HR = heart rate, IDd = inner dimension at diastole, LV = left ventricular, Max = maximal, MR = mitral regurgitation, P = pressure, RA = right atrial, RV = right ventricular, TIC = tachycardia-induced cardiomyopathy, TR = tricuspid regurgitation, TV = tricuspid valve.

*$p < 0.05$ vs. CTL, † $p < 0.05$ vs TIC (baseline).

*Figure 1b*). These increases in area appeared to be driven by a significant increase in width (p=0.023), and a near significant increase in height (p=0.070) (*Figure 1c–d*). Furthermore, the anterior leaflet areas showed no correlation with animal weight in either animal groups (CTL: r = 0.239, p = 0.253, TIC: r = 0.077, p = 0.349) (*Figure 1—figure supplement 1*). In summary, TIC anterior leaflet areas were 130% larger when compared to CTL anterior leaflets, ostensibly driven by an increase in leaflet width.

## Anterior leaflets thicken with TIC

Based on previous reports of ovine mitral valve leaflet thickening in mitral regurgitation (*Dal-Bianco et al., 2009*; *Dal-Bianco et al., 2016*), we hypothesized TIC anterior leaflets to be thicker than CTL anterior leaflets. To test this hypothesis, we measured anterior leaflet thickness from fixed radial strips of tissue (*Figure 2a*). In TIC and CTL animals, the mean thickness of all anterior leaflets decreased from the near-annulus region (thickest) to the free edge (thinnest) (*Figure 2b*). Furthermore, the mean thickness of TIC anterior leaflets was consistently larger than CTL anterior leaflets in all three regions. When summarized into a single average thickness, we found a significant increase (140%) in TIC anterior leaflet thickness (p=0.006, *Figure 2c*). When analyzed by region, the increases in thickness were most strongly driven by a significant increase in the free edge region (p=0.003, *Figure 2d*). In summary, TIC anterior leaflets were 140% thicker when compared to CTL anterior leaflets, mostly driven by increases in the free edge thickness.

## Increased metabolic and regulatory proteins in TIC anterior leaflets

We utilized proteomics to survey protein-level expression within the TIC anterior leaflets to infer which biological processes were up- or down-regulated in these tissues. To this end, we successfully identified 247 differentially expressed proteins between CTL and TIC anterior leaflets (*Appendix 1—figure 1*).

Refer to *Supplementary file 1* for a complete list of FASTA headers, gene names, family information, and expression levels for each of the 247 proteins identified. To better identify the molecular

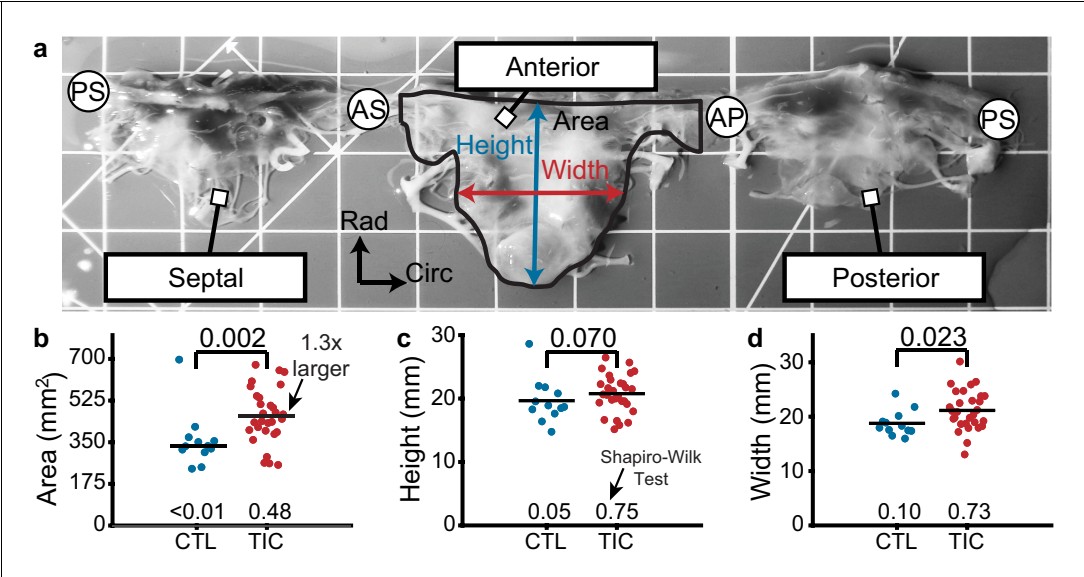

**Figure 1.** Tricuspid valve anterior leaflet area increases in sheep with tachycardia-induced cardiomyopathy (TIC). (a) An ovine tricuspid valve separated at the posterior-septal commissure (PS). Located between the antero-septal (AS) and antero-posterior (AP) commissures, the anterior leaflet and its measured area (black), height (blue), and width (red) are shown. Grid scale = 1 cm. (b–d) Comparisons between control (CTL, blue, n = 12) and TIC (red, n = 29) anterior leaflet (b) area, (c) height, and (d) width. Black bars represent data mean if normal, and data median if non-normal as determined by Shapiro-Wilk test: p-values below data. Values above data represent p-values from Student's t-test or Wilcoxon Rank-Sum test, as appropriate. The online version of this article includes the following figure supplement(s) for figure 1:

**Figure supplement 1.** Tricuspid valve anterior leaflet area change does not correlate with animal weight.

functions of the significant proteins, we built an interactome for easier visualization and interpretation of protein cluster interactions (*Appendix 1—figure 2*). Compared to CTL, TIC anterior leaflets overexpressed metabolic proteins (ENO1, ESD, ALDOA, ALDOC, ASPH, UGP2, AK1, PGK1, PGK2, PKM, LDHA, PGD, among others), serpins (A1, A5, C1, D1, and others), apolipoproteins (A1 and B), proteins related to matrix remodeling, such as COL6A3, which is tied to the alpha-chain of a type VI collagen, and FN1, GPC4, and PGLYRP1, all proteins involved in synthesis and regulation of glycans. Additionally, we found upregulation in specific proteins related to the mesenchymal phenotype, for example CD14, CD163, FERMT2, PLS3, and S100A1. Finally, we found that coagulation factor F9 was highly upregulated, along with a number of complement proteins (C2, C5, C6, and others), indicating the increased production of pro-inflammatory mediators downstream. We further classified the 247 differentially expressed proteins according to their primary families and gene ontologies (i.e., protein classes, molecular functions, cellular component, and biological process). We found that these 247 differentially expressed proteins most often belonged to protein classes of metabolite interconversion enzymes, protein binding modulators, and protein modifying enzymes. Furthermore, these 247 proteins were most associated with the molecular functions of catalytic, binding, and regulatory activity. The 247 proteins' most common cell component were organelles, protein complexes, or membranes. Finally, the most common biological processes for these 247 proteins were metabolic, biological regulation or biogenesis (e.g., histones, ribosomal, and membrane proteins). In summary, we found 247 differentially expressed proteins in TIC anterior leaflets when compared to CTL anterior leaflets, including many proteins suggestive of increased metabolic and regulatory processes.

## Expression of remodeling-associated cellular markers in TIC

We used marker expression in immunohistochemistry as a tool to determine regional snapshots of cell scale activity, cell phenotype, cell activation, or cell differentiation. Based on our hypothesis that anterior leaflet tissue remodels in disease, we investigated the presence of four cellular markers frequently associated with remodeling via immunohistochemistry: (i) α-smooth muscle actin (αSMA), (ii) Ki67, (iii) matrix metalloproteinase 13 (MMP13), and (iv) transforming growth factor β1 (TGF-β1)

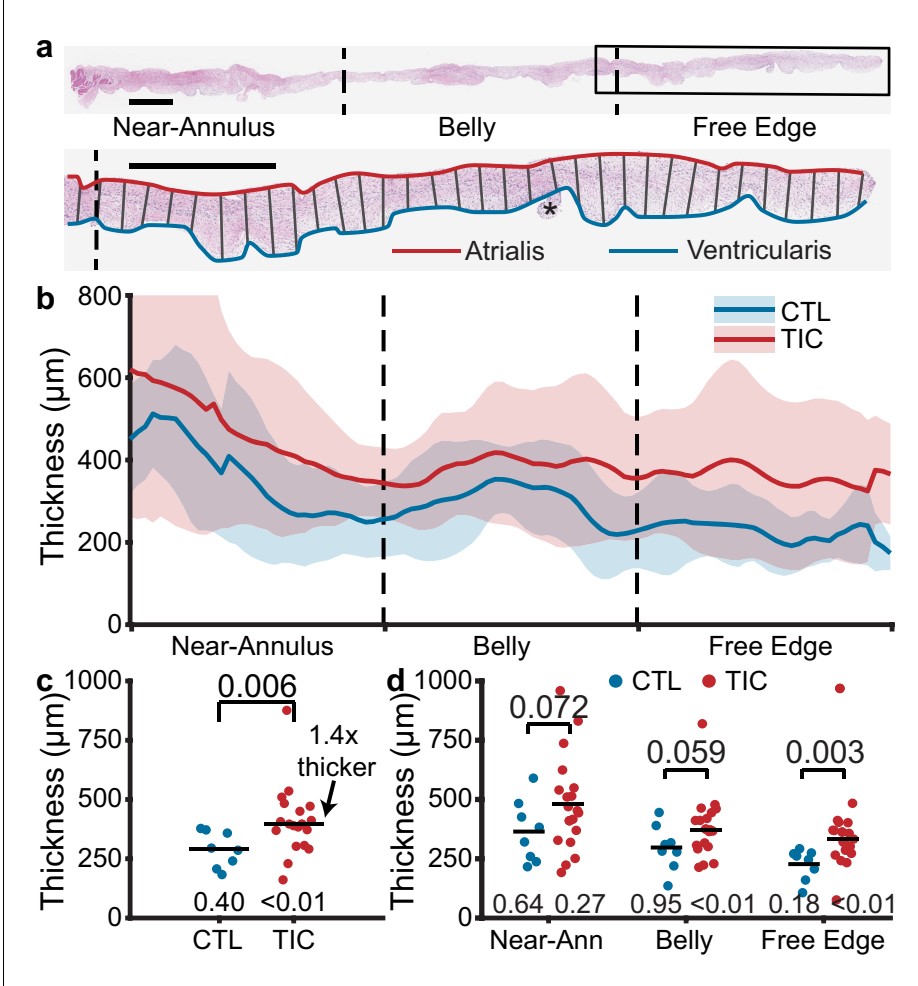

**Figure 2.** Anterior leaflet thickness increases in tachycardia-induced cardiomyopathy (TIC) are primarily driven by free edge thickening. (a) (top) Representative fixed radial tissue strip approximated into regions of near-annulus, belly, and free edge. Inscribed box (black) is magnified (below) to show thickness measurement between atrialis (red) and ventricularis (blue) splines. Chordae tendineae (*) were excluded manually. Scale bars = 1 mm. (b) Profiles of control (CTL, blue, n = 8) and TIC (red, n = 19) anterior leaflet thickness approximated into equal-third regions of near-annulus, belly, and free edge. Pictured are mean (solid) +/- 1 standard deviation (shaded). (c–d) Thickness comparisons between CTL and TIC groups when data is (c) pooled across all regions and (d) pooled within regions. Black bars represent data mean if normal, and data median if non-normal as determined by Shapiro-Wilk's test: p-values below data. Values above data represent p-values from Student's t-test or Wilcoxon Rank-Sum test, as appropriate.

(*Dal-Bianco et al., 2009*; *Stephens et al., 2009*; *Dal-Bianco et al., 2016*; *Stephens et al., 2008*). We found an increase in TIC αSMA expression, indicative of cellular activation, mostly in the atrialis of near-annulus and belly regions (*Figure 3a*). Furthermore, Ki67, a marker for cell proliferation, was increased for TIC in the atrialis of the belly region and in much of the free edge region (*Figure 3b*). However, we observed only marginal changes in TIC cell nuclei density in the near-annulus (increased) and free edge (decreased) regions (*Figure 3—figure supplement 1*). Additionally, MMP13, a collagenase, was widely increased for TIC in both near-annulus and belly regions (*Figure 3c*). Lastly, TGF-β1 also increased primarily in the belly region of TIC samples (*Figure 3d*). It should be noted that CTL tissues also expressed these markers, albeit to a lesser degree, likely suggesting their involvement in maintaining tissue homeostasis. Also note that, similarly to other data presented herein, immunohistochemistry stains showed heterogeneity among TIC subjects. That is, some TIC leaflets showed more positive staining than others. Overall, we observed regional increases in cellular markers associated with tissue remodeling processes.

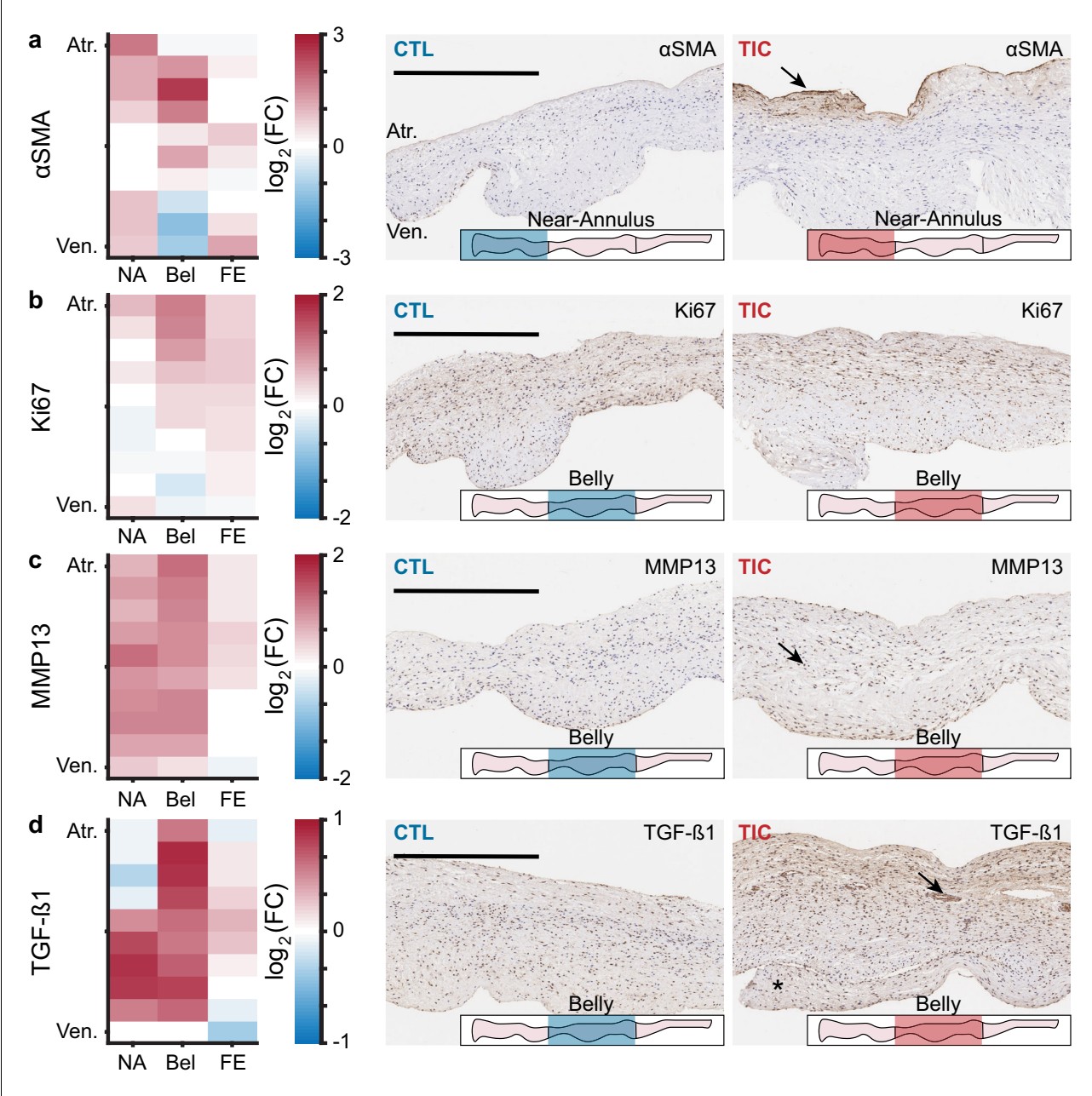

**Figure 3.** Remodeling cellular markers present in tarchycardia-induced cardiomyopathy (TIC) tissues. (a–d) Heat maps (left) showing regional expression of (a) alpha smooth muscle actin (αSMA), (b) Ki67, (c) matrix metalloproteinase 13 (MMP13), (d) transforming growth factor beta 1 (TGF-β1). Heat maps are separated by regions in radial (near-annulus (NA), belly (Bel), and free edge (FE)) and thickness (atrialis (Atr.), and ventricularis (Ven.)) axes. Fold change (FC) between control (CTL, n = 6) and TIC (n = 6) was determined by the ratio of positively stained pixel percentage between TIC and CTL. Color map indicates the logarithm base 2 of the fold change, interpreted as (positive, red): TIC expression is higher than CTL, (0, white): TIC and CTL expression are approximately equal, and (negative, blue): TIC expression is less than CTL. Representative images of CTL (middle) and TIC (right) are shown with atrialis surface upward. Black arrows indicate (a) increased TIC αSMA expression near the atrialis, (c) increase in TIC positively stained nuclei for MMP13, and (d) 'pocket' may be a neo-microvessel. Note, positive TGF-β1 staining in this pocket may be related to angiogenic signaling. Asterisk (*) denotes chordae tendineae excluded from analysis. Scale bars = 500 μm.

The online version of this article includes the following figure supplement(s) for figure 3:

**Figure supplement 1.** Marginal changes in TIC cellular nuclei density.

## Collagen content increases in the anterior leaflets in TIC

In the ovine mitral valve leaflets, collagen content increases with mitral regurgitation (*Stephens et al., 2009*; *Stephens et al., 2008*). Based on these results, we hypothesized that the TIC anterior leaflets would have an increased collagen content when compared to CTL anterior leaflets. To test our hypothesis, we quantified the wet tissue collagen content in the near-annulus, belly, and free edge regions of tissue. In both groups, we found collagen content decreased from the near-annulus region to the free edge (*Figure 4a*). Importantly, in all three regions, the mean collagen content was larger in TIC subjects than in CTL subjects. The strongest increase was in the free edge region (~150%, p=0.006) and the near-annulus region (~135%, p=0.023), but we failed to find a significant increase in the belly region (~125%, p=0.058). When we pooled all regions within subjects, we found a significant 140% increase in collagen content in TIC anterior leaflets (p=0.016, *Figure 4b*). We also tested if collagen content would positively correlate with TR severity, as collagen may stiffen and reduce the leaflet's range of motion. Here, we found a strong positive (r = 0.650) and significant (p=0.015) correlation (*Figure 4—figure supplement 1*). In summary, TIC anterior leaflets had 140% more collagen than CTL anterior leaflets, mostly driven by increases in the free edge and near-annulus content.

## Anterior leaflets stiffen in TIC

Based on our hypothesis that TIC anterior leaflets remodel in TR, we further hypothesized that this remodeling would result in altered mechanical properties. Specifically, we expected a stiffer and more isotropic behavior (*Grande-Allen et al., 2005b*). To mechanically characterize the tissue, we tested anterior leaflets under planar biaxial tension. Under this loading, all tissues exhibited classic J-shaped loading curves, consistent with other biological collagenous tissues (*Figure 5a*; *Sacks, 2000*). In detail, the transition stretches and degrees of anisotropy of the curves remained mostly consistent between CTL and TIC subjects, overall behaving stiffer in the circumferential direction than in the radial direction (i.e., degree of anisotropy <1) (*Figure 5—figure supplement 1*). However, our major finding from mechanical characterization was a significant 130% (p=0.006) increase in TIC anterior leaflet stiffness at large stretches (calf stiffness) in the radial direction. (*Figure 5b*). Additionally, we found that TIC anterior leaflets had significantly increased circumferential stiffnesses at small stretches (toe stiffness) (p=0.008, *Figure 5c*). In summary, TIC anterior

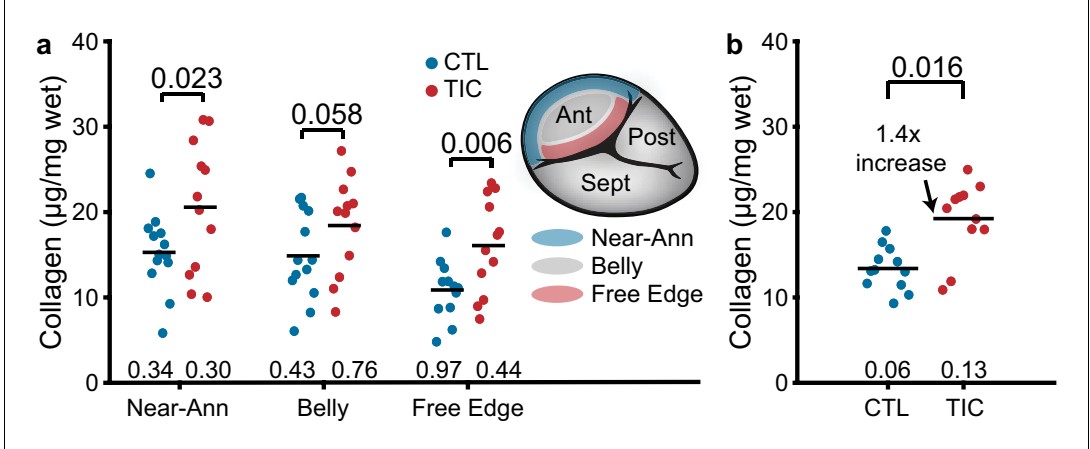

**Figure 4.** Increased collagen content in tachycardia-induced cardiomyopathy (TIC) anterior leaflets is primarily driven by increases in the near-annulus and free edge. (a) Wet weight collagen content from quantitative collagen assay comparisons between control (CTL, blue, n = 12) and TIC (red, n = 12) groups with tissue samples from near-annulus, belly, and free edge regions with inscribed region visualization on tricuspid valve anterior leaflet, and (b) when regions are averaged across subjects. Black bars represent data mean if normal, and data median if non-normal as determined by Shapiro-Wilk test: p-values below data. Values above data represent p-values from Student's t-test.

The online version of this article includes the following figure supplement(s) for figure 4:

**Figure supplement 1.** Collagen content positively correlates with clinical echocardiographic metric of tricuspid regurgitation (TR) severity.

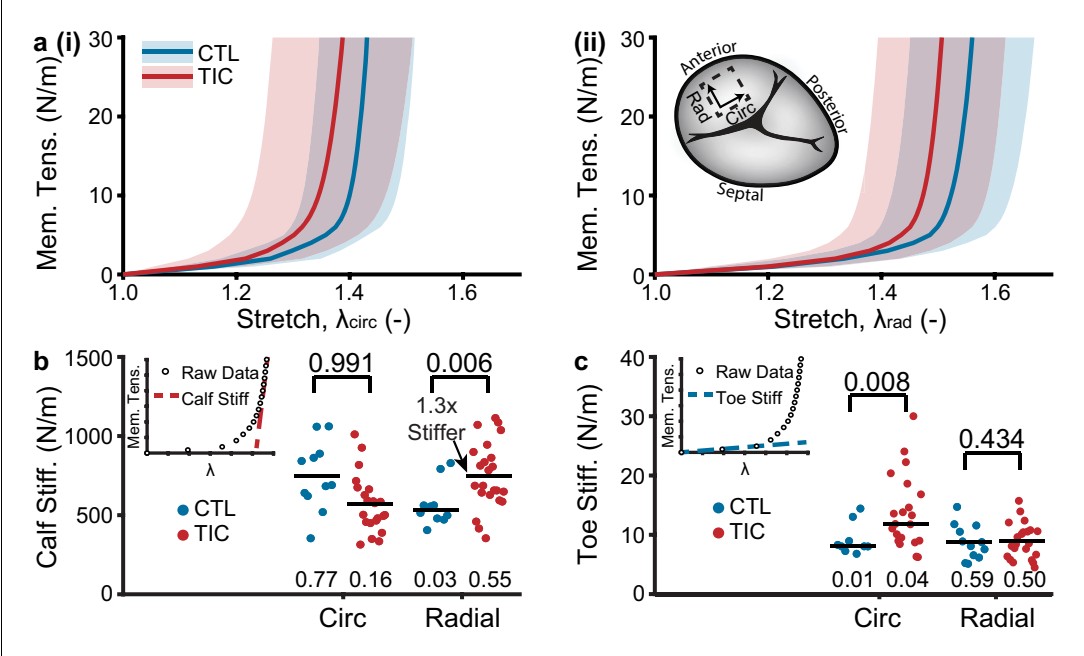

**Figure 5.** Tachycardia-induced cardiomyopathy (TIC) anterior leaflets are stiffer in circumferential (circ) and radial (rad) directions. (a) Control (CTL, blue, n = 11) and TIC (red, n = 23) membrane tension (Mem. Tens.) vs. stretch average curves (solid) with standard deviation (shaded) in (i) circumferential and (ii) radial directions. Inset in (a) is a visualization of where biaxial samples were acquired (dotted line) from anterior leaflets. (b–c) Comparisons of the (b) stiffness at large stretches (calf stiffness) and (c) stiffness at small stretches (toe stiffness) in circumferential and radial directions. Inset in (b) and (c) is the definition of calf stiffness (red, dashed) and toe (blue, dashed) we use to characterize a nonlinear material stiffness. Black bars represent data mean if normal and data median if non-normal as determined by Shapiro-Wilk test: p-values below data. Values above data represent p-values from Student's t-test or Wilcoxon Rank-Sum test, as appropriate.

The online version of this article includes the following figure supplement(s) for figure 5:

**Figure supplement 1.** Biaxial curves between control (CTL, blue, n = 11) and tachycardia-induced cardiomyopathy (TIC, red, n = 23) exhibited comparable anisotropies.

leaflets increased in stiffness. They increased predominantly in radial directions at higher stretches, and circumferential directions at lower stretches.

## Altered collagen fiber dispersion in the TIC anterior leaflet atrialis

To investigate the microstructure of anterior leaflets as the intermediary between tissue scale changes and mechanical properties, we used two-photon microscopy. Specifically, we visualized the collagen and cell nuclei distributions and orientations throughout the entire anterior leaflet thickness. In our analysis, we found that throughout the tissue depth, the mean collagen orientation remained mostly circumferential in CTL and TIC animals, with no apparent changes in mean direction among groups (*Figure 6a*). However, we noted an increase in collagen fiber dispersion between CTL and TIC (indicated by a larger concentration parameter κ). In other words, in TIC animals, more fibers were oriented in the radial direction than in the CTL subjects. When separating our analysis into three depth regions we found a statistically significant increase in fiber dispersion in the 0–33% depth, nearest the atrialis, (p=0.017) in TIC subjects (*Figure 6b*). Similar analyses in cell nuclei for nuclear orientation, nuclear aspect ratio (NAR), and circularity showed no notable differences between CTL and TIC cell nuclei (*Figure 6—figure supplement 1*). In summary, we observed that collagen fibers in TIC anterior leaflets were more radially dispersed than CTL anterior leaflets near the atrialis.

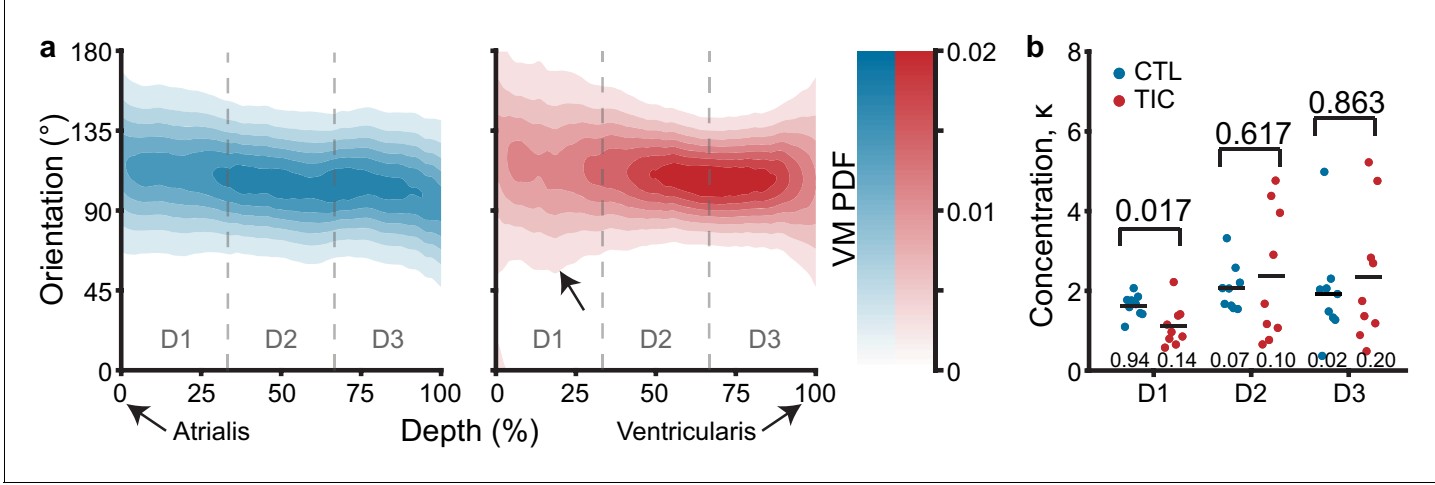

**Figure 6.** Through-depth collagen microstructure reveals similar mean orientations and regional concentration changes throughout depth between control (CTL) and tachycardia-induced cardiomyopathy (TIC). (a) Heat map visualizations of von Mises probability distribution functions (VM PDF) fit to the collagen fiber orientation histograms of two-photon acquired images throughout the entire depth (0% - Atrialis surface, 100% - Ventricularis surface) of averaged CTL (blue, n = 9) and TIC (red, n = 9) tissue samples. Orientations of 90° align circumferentially, while 0°/180° align radially. We observed qualitative regional concentration (i.e., heat map width) differences (arrow), which we quantified by (b) averaging VM concentration parameter, κ, across depth regions D1, D2, and D3, by subject. Black bars represent data mean if normal, and data median if non-normal as determined by Shapiro-Wilk test: p-values below data. Values above data represent p-values from Student's t-test, Welch's t-test or Wilcoxon Rank-Sum, as appropriate. The online version of this article includes the following figure supplement(s) for figure 6:

**Figure supplement 1.** Through-depth nuclei microstructure reveals similar nuclei orientation, nuclear aspect ratio (NAR) and circularity throughout depth between control (blue, n = 9) and tachycardia-induced cardiomyopathy (red, n = 9).

## Discussion

We investigated the tricuspid valve's propensity to (mal)adapt to TR, which was motivated by research on mitral valve (mal)adaptation to physiological (*Pierlot et al., 2015*) and pathological stimuli (*Dal-Bianco et al., 2016*; *Levine et al., 2015*; *Stephens et al., 2008*; *Beaudoin et al., 2017*). In the latter setting, we and others have shown that functional mitral regurgitation is not so functional after all (*Gillam, 2008*; *Rausch et al., 2012*; *Dal-Bianco et al., 2009*; *Stephens et al., 2009*; *Grande-Allen et al., 2005a*). To date, we have a preliminary, theoretical framework to link mitral leaflet (mal)adaptation to functional mitral regurgitation on multiple scales. We think that pathological mitral leaflet strain following ventricular remodeling and/or cellular signaling following ischemia initiate a (mal)adaptive cascade that activates quiescent, valvular interstitial cells and leads to the transdifferentiation of native endothelial cells into myofibroblast-like synthetic cells (*Levine et al., 2015*; *Ayoub et al., 2016*; *Ayoub et al., 2017*). Activation and transdifferentiation of native cell populations subsequently increase collagen turnover, and lead to collagen reorganization, tissue thickening and stiffening (*Dal-Bianco et al., 2009*; *Stephens et al., 2009*; *Grande-Allen et al., 2005b*). In turn, these tissue changes may lead to loss in coaptation competence and thus render the leaflets not passive bystanders but active contributors to functional mitral regurgitation. Our current study was inspired by those findings and sought to determine whether the tricuspid valve, too, (mal)adapts on all functional scales. To this end, we present our findings on tricuspid valve (mal)adaptation from the perspective of four different scales: protein scale, cell scale, matrix scale, and tissue scale.

On the protein scale, we have identified 247 differentially expressed proteins in TIC anterior leaflets, which we matched to several protein clusters. The increased metabolic protein expressions we observed may suggest increased energetic demands and a highly active response from valvular cells to TIC. This energy may be necessary for the synthesis of apolipoproteins, serpins and complement proteins, and matrix proteins, which all increased in expression. Increases in expression levels of such proteins have been observed in calcific (*Novaro et al., 2003*) and stenotic (*O'Brien et al., 1996*; *Schlotter et al., 2018*) aortic valves. Such increases may lead to imbalances in lipid

metabolism, increased inhibition of proteases such as antithrombins (*Janciauskiene, 2001*; *Sanrattana et al., 2019*) with consequent regulation of complement activation (*Beinrohr et al., 2011*; *Chakraborti et al., 2000*). Overall, the proteomic data indicate that anterior leaflet cells actively respond to TIC.

On the cell scale, using immunohistochemistry we have shown that expression of markers αSMA, Ki67, MMP13 and TGF-β1 were increased in TIC. As a marker for valvular interstitial cell (VIC) activation, increased αSMA suggests a fibrotic response and matrix remodeling, (*Liu et al., 2007*) consistent with our findings of increased collagen content. Conjunctively, increased MMP13 highlights extracellular turn-over activity through collagenases. While an increased Ki67 expression indicates cellular proliferation, further cellular recruitment from endothelial-to-mesenchymal transition (EndoMT) processes is likely with increased TGF-β1, as TGF-β is a mediator of EndoMT (*Wylie-Sears et al., 2014*) and VIC activation (*Walker et al., 2004*) in heart valves. Despite evidence of cellular proliferation and recruitment, cellular density of our tissues remained mostly similar, perhaps due to a balance by the increase in leaflet volume (i.e., area and thickness increase) we observed. Furthermore, these cell marker changes were regionally specific suggesting a heterogeneous cellular response, perhaps due to varying regional stimuli.

On the matrix scale, we found strong evidence of increased collagen content as well as an increase in collagen dispersion, that is disorganization, near the atrialis surface. The collagen increase may be attributed to the increased fibrotic activity on the cellular scale. Regionally, we observed that collagen increases were most prominent in the near-annulus and free edge regions. Considering the proximity of the near-annulus leaflet tissue to the annulus and the abundance of chordal attachments near the leaflet free edge, we propose that annular dilation and chordal tethering resulting from ventricular remodeling mechanically deformed these regions of leaflet beyond normal physiological levels, inciting an increased profibrotic response by cells in these areas in an attempt to maintain mechanobiological homeostasis (*Gupta and Grande-Allen, 2006*). Secondarily, we observed a more radially dispersed collagen orientation in the atrialis region of the leaflets. Although this may be evidence of radial collagen deposition due to increased radial leaflet stresses from chordal tethering, it is also possible that altered shear stresses from regurgitant blood flow across the atrialis surface induced this microstructural adaptation.

At the tissue scale, we observed leaflet area increase, thickening, and calf region stiffening in the radial direction. Importantly, as opposed to in vivo measurements of area increase that are taken under mechanical stress (due to chordal attachments, etc.), our measurements were taken in a mechanically stress-free state. Thus, area increase is entirely due to growth as opposed to elastic stretch. Growth was driven by increases in both width and height (although the latter was not significant). Additionally, we found that tissue calf-region stiffness increased in the radial direction and seemingly reduced in the circumferential direction. These changes may be direct responses to annular dilation and papillary muscle displacement in TIC animals that respectively alter circumferential and radial strain. In turn, altered strain may disrupt the mechanobiological equilibrium in those directions, eliciting increased collagen deposition (radial) or degradation (circumferential) in an attempt to reestablish homeostasis (*Gupta and Grande-Allen, 2006*). Our stiffness results agree with our microstructural observations of increased collagen fiber dispersion in the radial direction and may suggest radial stiffening is an adaptive mechanism of the anterior leaflet. Interestingly, we did observe a significantly stiffer circumferential toe region, which may suggest a role in remodeling for elastin, which is often attributed with controlling the toe-region mechanical behavior of heart valve tissue. Additionally, tissue thickness increase may be a direct consequence of upregulated collagen deposition. However, it cannot be ruled out that thickness increase may also be due to inflammation-induced tissue swelling.

We set out to identify whether the tricuspid valve, like the mitral valve, (mal)adapts. We provide significant evidence in support of the tricuspid valve's propensity to (mal)adapt on several scales. It does so in a qualitatively very similar way to the mitral valve. Although, direct comparison should be done with caution as our current model differs from those of mitral valve (mal)adaptation. Specifically, we found similar tissue-level, matrix-level, and cell-level changes. The most significant difference being that we did not find strong evidence for EndoMT. Specifically, we did not find CD31+ staining cells co-localized with αSMA, as reported by *Bartko et al., 2017*. Similarly, we found no significant upregulation of proteins related to EndoMT in our proteomic analysis. While the reasons

could be multi-fold, one possible explanation could be that EndoMT may have occurred before or after our observational period.

These findings are clinically significant. Similar findings to ours in the mitral valve have inspired the study of pharmacological treatment strategies to reduce leaflet maladaptation, that is detrimental tissue effects, while maintaining its adaptive, that is beneficial, growth. Specifically, the use of Losartan - an angiotensin II type one receptor antagonist that indirectly blocks the TGF-β1 phosphorylation of ERK in EndoMT (*Wylie-Sears et al., 2014*) - resulted in reduced mitral valve leaflet thickness, EndoMT, cellular activation, cell recruitment, and collagen deposition, while preserving leaflet area growth (*Bartko et al., 2017*). While much remains to be understood, we hope that we have taken a first step toward similar studies that may support pharmacological treatment of TR surgery and/or intervention. Interestingly, recent work has demonstrated that not only disease, but also surgical repair may initiate mitral valve tissue (mal)adaptation (*Sielicka et al., 2018*). It is possible that the tricuspid valve, may also (mal)adapt to repair. If true, our current and future work, should focus on delineating the potentially differing mechanistic origins of tricuspid valve adaptation and maladaptation. Through such studies, we may learn how to promote the former, while preventing the latter. For example, promoting adaptive area increase of the valve could improve interventional efficacy by facilitating better leaflet coaptation. Similarly, avoiding maladaptive stiffening and thickening could prevent post-surgical repair failure. Such strategies could have broad impact by enabling optimization of both surgical repair techniques and interventional approaches, such as transcatheter annuloplasty (*Kuwata et al., 2017*) or leaflet clipping (*Fam et al., 2018*).

Naturally, our study is subject to several limitations. Regarding our animal model, we chose a tachycardia-induced biventricular heart failure model over a pulmonary hypertension model as it better represents the clinical setting of TR. That is, few patients present with isolated TR. In fact, 85–90% of severe TR cases are functional, resulting predominantly from left-sided heart disease (*Mangieri et al., 2017*). Our model, with which we have years of experience, demonstrates all salient features of biventricular disease such as reduced left and right ventricular ejection fraction, dilated ventricles, mitral and tricuspid annular dilation, and regurgitation in both valves (*Malinowski et al., 2017*). However, our ovine model develops biventricular heart failure within weeks as opposed to years as in patients. Thus, we are likely only reporting on the early response of the tricuspid leaflets to disease. As we fail to capture the late response, we ask the reader to extrapolate our findings with caution. Future studies should extend this time window. The complexity of our model is likely also reflected in our inability to find clear correlative relationships between measures of (mal)adaptation and potential (mal)adaptive stimuli, such as severity of TR or measures of right ventricular dilation. However, as our intent was to evaluate the valve's propensity to growth and remodeling, that is (mal)adapt, in the presence of functional TR, we do not see this as a significant limitation. Nonetheless, future studies should aim to isolate the underlying mechanisms through use of other models. Finally, we focused our study only on the anterior leaflet, while the posterior and septal leaflets remain unreported on. With recent publications highlighting unique mechanical and microstructural properties between tricuspid valve leaflets (*Meador et al., 2020a*) as well as heterogeneous properties within leaflets, (*Laurence et al., 2019*; *Kramer et al., 2019*) we suggest that future studies investigate the heterogeneity of the (mal)adaptive response in the entire tricuspid valve. Although limitations remain, they provide clear future directions for this line of research.

In conclusion, we used an ovine tachycardia-induced biventricular heart failure model to reveal that the anterior tricuspid valve leaflet (mal)adapted on multiple scales. Namely, we observed in the leaflets of these animals: (i) an active metabolic protein response suggestive of increased energy expenditure, (ii) cells native to the anterior leaflet expressed markers indicative of (mal)adaptation, (iii) upregulated collagen synthesis, (iv) reorganized collagen microstructure, (v) increased tissue thickness, (vi) increased leaflet area, and (vii) increased stiffness. As in the mitral valve, we submit that these multi-scale changes are linked. Specifically, we borrow from our understanding of the mitral valve and propose that, as pathological stimuli activate valvular interstitial cells, changes in cellular phenotype and cell proliferation likely emphasize inflammatory and profibrotic signaling pathways, which upregulate protein expression that initiates increased matrix turnover. As tissue-scale properties are dependent upon their micro-scale composition, the tissue consequently exhibits altered morphological and mechanical properties as a direct result of cellularscale activity.

# Materials and methods

### Key resources table

| Reagent type (species) or resource | Designation | Source or reference | Identifiers | Additional information |
|---|---|---|---|---|
| Biological sample (Ovis aries, male) | Dorset Sheep | Hunter Dorsets (http://www.hunterdorsets.com) | | |
| Antibody | (Rabbit polyclonal) anti- αSMA | Abcam | Cat#:Ab5694; RRID:AB_2223021 | IHC (1:600) |
| Antibody | (Rabbit polyclonal) anti- Ki67 | Thermo Scientific | Cat#:Rb1510-P0; RRID:AB_60158 | IHC (1:200) |
| Antibody | (Rabbit polyclonal) anti-MMP13 | Abcam | Cat#:Ab39012; RRID:AB_776416 | IHC (1:1000) |
| Antibody | (Rabbit polyclonal) anti-TGFβ−1 | Abcam | Cat#:Ab9758; RRID:AB_296604 | IHC (1:100) |
| Chemical compound, drug | Hoechst 33342 | Invitrogen | Cat#:H3570 | (5 µg/mL) |
| Commercial assay or kit | Total Collagen Assay Kit (Perchlorate-Free) | BioVision Incorporated | Cat#:K406 | |
| Commercial assay or kit | Pierce Microplate BCA Protein Assay Kit | Thermo Scientific | Cat#:23252 | |
| Software, algorithm | ImageJ (OrientationJ) | ImageJ (http://imagej.nih.gov/ij); PMID:21744269 | RRID:SCR_003070; RRID:SCR_014796 | |

## Animal model, medications, and procedures

All aspects of this research study were performed in accordance with the Principles of Laboratory Animal Care, formulated by the National Society for Medical Research, and the Guide for Care and Use of Laboratory Animals prepared by the National Academy of Science and published by the National Institutes of Health. Additionally, this protocol was developed, reviewed and performed in accordance with the approval of a local Institutional Animal Care and Use Committee.

The tachycardia-induced cardiomyopathy ovine model used in this study has been previously described in detail and validated as a reliable and repeatable model of biventricular heart failure in sheep (*Malinowski et al., 2017*). In short, we randomly assigned adult male Dorset sheep to either control (CTL, n = 17, 59.9 ± 4.6 kg) or disease (TIC, n = 33, 60.1 ± 5.3 kg) groups. Initially, all animals were acclimated for a 7 day period. In all animals, we monitored baseline (i.e. prior to pacing) ventricular function and valvular competence via epicardial echocardiography. We acquired all echocardiographic images with a 1.5–3.6 MHz transducer and Vivid S6 ultrasound machine (GE Healthcare, Chicago, IL). We used the American Society of Echocardiography criteria to assess valvular insufficiency. Using color flow and continuous wave Doppler, an experienced cardiologist graded TR as none (0), trace (0.5), mild (1), moderate (2), moderate-severe (3), or severe (4). We applied local anesthesia with 1% lidocaine for external right jugular intravenous (IV) catheter placement. We then initiated anesthesia (propofol 2–5 mg/kg IV), intubated, and mechanically ventilated each animal. We maintained general anesthesia via isoflurane inhalation (1%–2.5%) and fentanyl (5–20 µg/kg/minute). We monitored arterial blood pressure through an 18-gauge left carotid artery catheter (Teleflex, Morrisville, NC).

In TIC subjects, we sutured a monopolar pacing lead onto the lateral left ventricular wall, gaining access through a left lateral mini-thoracotomy approach (10–15 cm, fifth/sixth intercostal space) with a surgical incision approximated in standard fashion. Additionally, we infiltrated intercostal nerves with bupivacaine (0.25%). We exteriorized the lead through the thorax to a pacemaker (Consulta CRT-P, Medtronic, Minneapolis, MN). We stabilized the pacemaker in the subcutaneous pocket near the spine. Once the animal was breathing spontaneously, standing, and eating in the recovery area, we moved the animal to the pen. We provided prophylactic antibiotics preoperatively and postoperatively for 10 days (cefazolin 2 g IV q12H, and gentamicin 240 mg IV q24H). After 4–5 days of recovery, we initiated a progressive 180–260 beats per minute pacing protocol. During the pacing of TIC animals, we performed surveillance transthoracic echocardiography every three days to assess heart

failure progression. One hour prior to each echocardiography study, we paused pacing, which was resumed at the conclusion of each study. Once both moderate TR and left ventricular dysfunction (ejection fraction <30%) were present (19 ± 6 days), we performed a terminal epicardial echocardiograph.

During the terminal procedure of all animals, we acquired hemodynamic data over three full cardiac cycles via pressure transducers (PA4.5-X6; Konigsberg Instruments, Pasadena, CA) in the left and right ventricles via the apex, and the right atrium. We acquired cardiac pressure data and electrocardiographic recordings at 128 Hz over three cardiac cycles. We defined end-diastole as the time of the peak of the R-wave, and end-systole as the time of the maximum negative time derivative of left ventricular pressure. We euthanized all animals via sodium pentothal (100 mg/kg IV) and potassium chloride bolus (80 mEq IV). In CTL animals, the terminal procedure was performed after baseline epicardial echocardiograph. Lastly, we isolated the tricuspid valve from each animal.

For disclosure, we previously published the morphological, thickness, biaxial mechanics, and two-photon data for n = 6 CTL sheep included in this study (*Meador et al., 2020a*). However, these data were all supplemented with additional subjects for this study, as the focus of that study was to report our findings in only CTL matched subjects. Additionally, to maximize utilization of each animal model, some TIC subjects were also used in a previously published study, unrelated to the leaflet (mal)adaptive response (*Jazwiec et al., 2019*).

## Morphology and storage

Immediately after tricuspid valve isolation, we cut open the valve leaflet complex by separating the leaflets at the posterior-septal commissure to enable valve unfolding. We floated the tricuspid valve, atrialis side up in 1xPBS and orthogonally photographed the tricuspid valve leaflets on a calibrated grid. Using these photographs, an experienced cardiac surgeon identified the commissural points at which leaflets were separated and, using custom MATLAB code, we calculated the anterior leaflet area, major cusp width, and major cusp height (*Figure 1a*). All pixel measurements were translated to length metrics by means of the calibrated grid. We then cryogenically stored the tissue at −80°C in a 9:1 ratio of DMEM (VWR L0101-0500, Radnor, PA, USA):DMSO (VWR BDH1115-1LP) with protease inhibitor (ThermoFisher, A32953, Weltham, MA, USA) until further testing. Once ready to be tested, we rapidly thawed the vials in room temperature water and removed all chordae tendineae from the ventricularis surface of the anterior leaflets prior to tissue sample collection for testing.

## High-throughput analysis of protein expression

We used tricuspid valve anterior leaflet fragments without consideration of localization/region for high-throughput mass spectrometry analyses. We finely minced the tissues and lysed cells at 4°C in a buffering solution containing 50 mmol/L HEPES (pH = 7.4) supplemented with 150 mmol/L sodium chloride, 2 mmol/L dithiothreitol, 1% IGEPAL and a protease inhibitor cocktail (Active Motif, 100546). We incubated a total of six samples (n = 3 CTL and n = 3 TIC) with mild agitation at 4°C for 2 hr. We determined the total protein concentration of each sample by bicinchoninic acid assay (Thermo, PI-23252). After quantification, we ethanol precipitated 50 μg of protein from each sample. After precipitation, we re-dissolved the proteins in Laemmli buffer for one-dimensional gel electrophoresis, Coomassie blue staining and in-gel digestion, following a previously published protocol (*Shevchenko et al., 2006*). We dehydrated the gel pieces successively in 50% and 100% acetonitrile, and digested the proteins overnight in 0.1 μg sequencing-grade trypsin contained in 50 μL total volume. Finally, we extracted the peptides twice in 50% acetonitrile with 0.1% formic acid, which we dried to completion in a vacuum centrifuge.

We submitted the dried peptides to the University of Texas at Austin CBRS Biological Mass Spectrometry Facility for protein identification by liquid chromatography-tandem mass spectrometry (LC-MS/MS) using the Dionex Ultimate 3000 RSLCnano LC coupled to the Thermo Orbitrap Fusion (*Lee et al., 2013*). Prior to HPLC separation, we desalted the peptides using Millipore U-C18 ZipTip pipette tips following the manufacturer's protocol. We performed a C18 trap column, followed by a 75 μm I.D. x 25 cm long analytical column packed with C18 3 μm material (Thermo Acclaim PepMap 100) running a gradient from 5–35% acetonitrile. With a Fourier Transform Mass Spectrometry resolution of 120,000, and a 3 s cycle time, we acquired MS/MS in HCD ion trap mode. We uploaded the LC−MS/MS mzXML files to MaxQuant (Max Planck Institute of Biochemistry, Germany) and

searched against a protein database of Ovis aries downloaded from UniProt. We used default search parameters in MaxQuant for label-free quantification with a 1% false discovery rate. We selected only the proteins which had at least two replicates of intensity values across all treatment groups for the remaining analyses. With this, we confidently identified a total of 2457 proteins. We normalized intensity values per total protein content, and we set a fold-change of two prior to analysis in the Multi-Experiment Viewer (MeV, TM4 Microarray Software Suite). Using the non-parametric Wilcoxon Rank-Sum statistical test to compare proteins in the two groups, we identified 247 significant differentially expressed proteins. We uploaded this list of proteins onto STRING v11 (Search Tool for the Retrieval of Interacting Genes/Proteins, STRING Consortium) for the generation of protein interaction maps. We further used the PANTHER (Protein ANalysis THrough Evolutionary Relationships, http://www.pantherdb.org/) classification system for the identification of Gene Ontologies (GO, http://www.geneontology.org). Refer to *Supplementary file 1* for all protein identifiers, protein family descriptions, and ontology classifications.

## Histology, thickness, and immunohistochemistry

We fixed radial strips (*Figure 2a*) of anterior leaflets from annulus to free edge in 10% Neutral Buffered Formalin for 24 hr, and transferred them to 70% ethanol for storage until histological processing. We shipped the samples to a histological service (HistoServ, Inc, Amaranth, MD) for embedding, sectioning (5 µm), and H and E staining. Using a light microscope (BX53 Upright Microscope, Olympus, Tokyo, Japan) with a 10x objective, we acquired and stitched full section images. Using custom MATLAB code, we fit splines to the atrialis and ventricularis surface of each section. We calculated the normal vectors along the atrialis spline, using these vectors to determine the distance between atrialis and ventricularis splines (*Figure 2a*). Along the length, we summarized the thickness measurements into three regions (i.e. near-annulus, belly, and free edge), defined as three equidistant regions along the arc length of the atrialis spline.

Additionally, the commercial histological service (HistoServ, Inc, Amaranth, MD) performed four immunohistochemistry stains on the same radial strips. These stains include commonly used markers for growth and remodeling processes in previous studies (*Dal-Bianco et al., 2009*; *Stephens et al., 2009*; *Dal-Bianco et al., 2016*; *Stephens et al., 2008*): (i) α-smooth muscle actin (α-SMA) (Abcam, ab5694, Cambridge, MA, US) as a marker for valvular interstitial cell (VIC) activation, (ii) Ki-67 (Thermo Scientific, rb-1510-P0, Waltham, MA, US) to determine cell proliferation, (iii) matrix metalloproteinase 13 (MMP13) (Abcam, ab39012, Cambridge, MA, US) to quantify collagenases, and (iv) transforming growth factor β1 (TGF-β1) (Abcam, ab9758, Cambridge, MA, US) as a key profibrotic factor. Using the light microscope with a 40x objective, we acquired and stitched full section images. Using custom MATLAB code, we fit splines to the atrialis and ventricularis surface of each section. Next, we excluded regions of annular muscle, as annular muscle often skewed positive results. Furthermore, we were specifically interested in the location of these markers as regions of tissue may be differentially active, due to regionally varying stimuli and composition (*Meador et al., 2020a*; *Laurence et al., 2019*; *Kramer et al., 2019*). Along three equidistant length regions of the atrialis spline, again representing near-annulus, belly, and free edge regions, we interpolated 10 thickness regions between the atrialis and ventricularis splines - altogether splitting each radial strip section into 30 regions. We then passed each region of the image into a custom validated MATLAB program which detects positive and total pixels in that region, from which we determined the normalized percentage of positive stain in that region.

## Quantitative collagen assay

We acquired the wet mass of tissue samples from annulus, belly, and free edge regions of anterior leaflets (*Figure 4a*). For every 10 mg of wet tissue mass, we added 100 µL dH20 for homogenization. We hydrolyzed 100 µL of homogenate in 100 µL of 10 N concentrated NaOH at 120˚C for 1 hr, after which we neutralized by adding 100 µL of 10 N concentrated HCl. After vortex mixing at 2000x g for 5 min, we transferred 10 µL of hydrolysate to each well in triplicate, which we allowed to evaporate to dryness on a 65˚C heating plate. We then followed the protocol provided with a total collagen assay kit (Biovision Inc, K406, Milpitas, CA, USA). From the quantitative assay, we measured the colorimetric absorbance at 560 nm with a spectrophotometer (Tecan, Infinite 200 Pro, Männedorf, Switzerland) which we interpolated from a collagen type I standard linear-fit curve ($R^2 = 0.99 \pm 0.01$).

Note: Due to tissue shortage, this experiment was supplemented with 7 CTL sheep (male, Dorset, 46 ± 6 kg) for a second cohort of control animals.

## Biaxial testing and analysis

We analyzed biaxial mechanics using methods previously described (*Meador et al., 2020a*). We isolated 7 x 7 mm square samples from the belly region of anterior leaflets, ensuring the major axes of the square aligned with the radial and circumferential directions of the leaflet (*Figure 5a*). On the atrialis surface, we applied an approximate 3 x 3 mm grid of four ink fiducial markers in the tissue center to enable strain tracking during testing. To establish a stress-free reference configuration, we photographed these fiducial markers while the tissue floated in 1xPBS on a calibrated grid. We mounted each sample on a biaxial testing device (Biotester, Cellscale, Waterloo, ON, Canada) with rakes, ensuring the radial and circumferential axes of the tissue aligned with the axial directions of the device. After submerging the mounted samples in 37°C 1xPBS, we initiated a 10 cycle preconditioning force-controlled protocol to 300 mN equibiaxially. After preconditioning we removed the tissue slack with a 10 mN preload. We tested the tissue at 300 mN equibiaxially for two cycles each, during which we recorded (5 Hz) rake-to-rake distances, circumferential and radial forces, and images of the fiducial markers to compute local tissue stretches of the final downstroke. All biaxial tests were within four hours of tissue thawing.

Using the coordinates of the fiducial markers on the tissue throughout testing, we calculated the deformation gradient tensor, $F$, with respect to the floating stress-free reference configuration. We then calculated the right Cauchy-Green deformation tensor, $C$, via $C = F^T F$, to acquire in-plane stretches throughout testing. Using the force data, we calculated the membrane tension as the force divided by the rake-to-rake distance in the orthogonal direction in the deformed configuration at each time point. In our analysis, we characterize these nonlinear J-shaped curves with four metrics: (1) toe stiffness, as the slope of the lower linear region of the curve, (2) calf stiffness, as the slope of the upper linear region (near 20 N/m membrane tension) (3) transition stretch, as the stretch at which the heel of the J-shaped curve is located (determined as the closest data point to the intersection of toe and calf stiffness lines), and (4) anisotropy index, as the ratio between the circumferential and radial stretches at 20 N/m.

## Two-photon microscopy

We analyzed the collagen microstructure and cell nuclei morphology using two-photon microscopy methods previously described (*Meador et al., 2020a*; *Meador et al., 2020b*). Briefly, post-mechanical testing, we counterstained the 7 mm x 7 mm belly tissue samples for cell nuclei (Thermo Fischer Scientific, Hoechst 33342, Waltham, MA, US) for 20 min. Afterward, we optically cleared the tissue in an optical clearing solution (Glycerol:DMSO:5xPBS, 50:30:20%) under sonication for 30 min to improve imaging depth. We imaged three centrally located 500 x 500 µm regions through the entire tissue thickness at 10 µm z-steps with a two-photon microscope (Bruker, Ultima IV, Billerica, MA, US) and 20x water immersion objective (Olympus, XLUMPLFLN, Center Valley, PA, US). We imaged with a coverslip on the atrialis surface to avoid imposed stresses, and the tissue on a foil-lined glass slide to visually verify full-thickness image acquisition. We utilized second harmonic generation (SHG) at an excitation wavelength of 900 nm and fluorescence at an excitation wavelength of 800 nm to image collagen and cell nuclei, respectively. We epicollected the emission signal of collagen and cell nuclei with a photomultiplier tube filter (460 ± 25 nm).

On each collagen image, we acquired coherency-based histograms of collagen fiber orientation by first normalizing the image histogram based on saturation and utilizing the ImageJ plugin OrientationJ on this processed image (*Rezakhaniha et al., 2012*). Across the three z-stacks of each sample, we averaged and interpolated the OrientationJ output data into a single z-stack of normalized histograms from atrialis to ventricularis surfaces. From there, we fit von Mises distributions to each histogram acquiring the parameters µ and κ representing the mean fiber angle and the fiber orientation concentration, respectively. Collectively, for every sample we imaged, we acquired a set of von Mises parameters representing mean fiber angles and fiber concentration at all depths. On each cell nuclei image, we acquired metrics of nuclear orientation, nuclear aspect ratio (NAR), and circularity by means of a custom MATLAB program which identifies individual nucleus contours in each image. With many nuclei in a single image, we acquire histograms of each of the three metrics at each

depth of the z-stack, which we process in the same way as above. For nuclear orientation, we used von Mises distribution fits, while on NAR and circularity we fit normal distributions with parameters μ and σ representing the mean and standard deviation, respectively.

## Statistical analyses

For all experimental data, we first performed Shapiro-Wilk tests to determine whether our data were normally distributed. Additionally, we tested whether the variances of our data sets were similar through an F-test. Under all disease versus control comparisons, a Student's t-test was used if the data were normally distributed and had similar variances. If these assumptions were not met, we used the Wilcoxon Rank-Sum Test or Welch's t-test, as appropriate. In correlations, if either variable failed the normality assumption or was ordinal in type, we used the nonparametric Spearman rank correlation. Otherwise, we used a Pearson correlation. For statistical comparisons and correlations we used either one-tailed or two-tailed tests, as appropriate, with our stated hypotheses. We defined a p-value less than 0.05 as significant. All statistical comparisons and correlations were implemented in MATLAB.

## Acknowledgements

Research reported in this publication was supported by the National Heart, Lung, And Blood Institute of the National Institutes of Health under Award Number F31HL145976 (WDM), the American Heart Association for their support under Award Number 18CDA34120028 (MKR), and internal grants from Meijer Heart and Vascular Institute at Spectrum Health. The content is solely the responsibility of the authors and does not necessarily represent the official views of the National Institutes of Health or the American Heart Association.

## Additional information

### Competing interests

Manuel K Rausch: Consultant for Edwards Lifesciences. The other authors declare that no competing interests exist.

### Funding

| Funder | Grant reference number | Author |
| --- | --- | --- |
| American Heart Association | 18CDA34120028 | Manuel Karl Rausch |
| National Institutes of Health | F31HL145976 | William D Meador |

The funders had no role in study design, data collection and interpretation, or the decision to submit the work for publication.

### Author contributions

William D Meador, Conceptualization, Data curation, Software, Formal analysis, Investigation, Methodology, Writing - original draft; Mrudang Mathur, Data curation, Software, Formal analysis; Gabriella P Sugerman, Carla MR Lacerda, Investigation, Methodology, Writing - original draft; Marcin Malinowski, Conceptualization, Investigation, Methodology, Writing - original draft; Tomasz Jazwiec, Xinmei Wang, Investigation, Methodology; Tomasz A Timek, Resources, Supervision, Funding acquisition, Writing - original draft; Manuel K Rausch, Conceptualization, Resources, Data curation, Supervision, Funding acquisition, Investigation, Methodology, Writing - original draft, Project administration

### Author ORCIDs

William D Meador https://orcid.org/0000-0001-5797-9553
Mrudang Mathur https://orcid.org/0000-0002-9273-5586
Gabriella P Sugerman https://orcid.org/0000-0001-9904-5257

Marcin Malinowski (iD) https://orcid.org/0000-0002-4526-2843
Manuel K Rausch (iD) https://orcid.org/0000-0003-1337-6472

### Ethics

Animal experimentation: All aspects of this research study were performed in accordance with the Principles of Laboratory Animal Care, formulated by the National Society for Medical Research, and the Guide for Care and Use of Laboratory Animals prepared by the National Academy of Science and published by the National Institutes of Health. Additionally, this protocol was developed, reviewed, and performed in accordance with the approval of the local Institutional Animal Care and Use Committee at Spectrum Health. The specific approval numbers are 2017-357, 2018-402, 2018-427, and 2018-439.

### Decision letter and Author response

Decision letter https://doi.org/10.7554/eLife.63855.sa1
Author response https://doi.org/10.7554/eLife.63855.sa2

## Additional files

### Supplementary files

• Supplementary file 1. (Sheet A) List of FASTA headers, gene names, family information and expression levels for each of the 247 proteins. (Sheet B) Protein families identified, and their molecular functions, biological processes, cellular components, and protein class. (Sheet C) Chart summary of protein families identified and their common roles.

• Transparent reporting form

### Data availability

The proteomic data are publicly available on ProteomeXchange (http://www.proteomexchange.org/) under accession number: PXD022719. The data to Figures 1-6 and their appendices are publicly available on the Texas Data Repository (https://doi.org/10.18738/T8/KBL2XE).

The following datasets were generated:

| Author(s) | Year | Dataset title | Dataset URL | Database and Identifier |
| --- | --- | --- | --- | --- |
| Meador WD, Rausch MK | 2020 | Tricuspid Leaflet Mechanics | https://doi.org/10.18738/T8/KBL2XE | Texas Data Repository, 10.18738/T8/KBL2XE |
| Lacerda CM, Meador WD, Rausch MK | 2020 | Tricuspid Leaflet Proteomics | http://proteomecentral.proteomexchange.org/cgi/GetDataset?ID=PXD022719 | ProteomeXchange, PXD022719 |

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

## Appendix 1

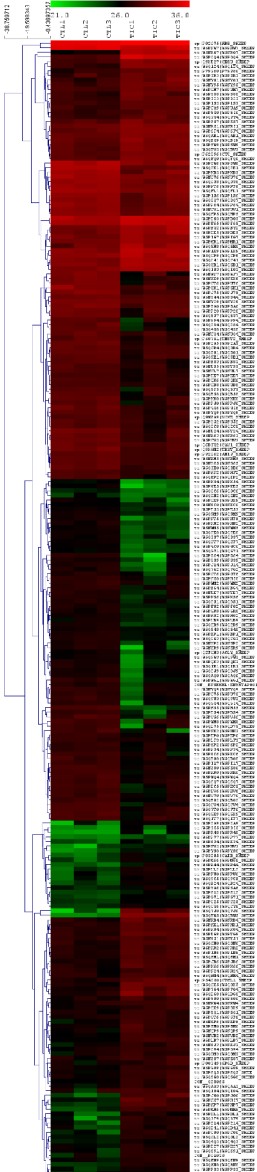

**Appendix 1—figure 1.** Proteomics revealed 247 differentially expressed proteins in tachycardia-induced cardiomyopathy (TIC) tricuspid anterior leaflets. Heat map showing expression of significant (p<0.05, fold change >2) differentially expressed proteins between control (CTL, left three columns, n = 3) and TIC (right three columns, n = 3) groups. Each column represents a biological replicate within each group. Each row represents a protein, as marked by their uniport identifiers. Red, black and green correspond to high, medium and low intensity values, respectively. Proteins are clustered according to similarities in expression patterns and total intensities.

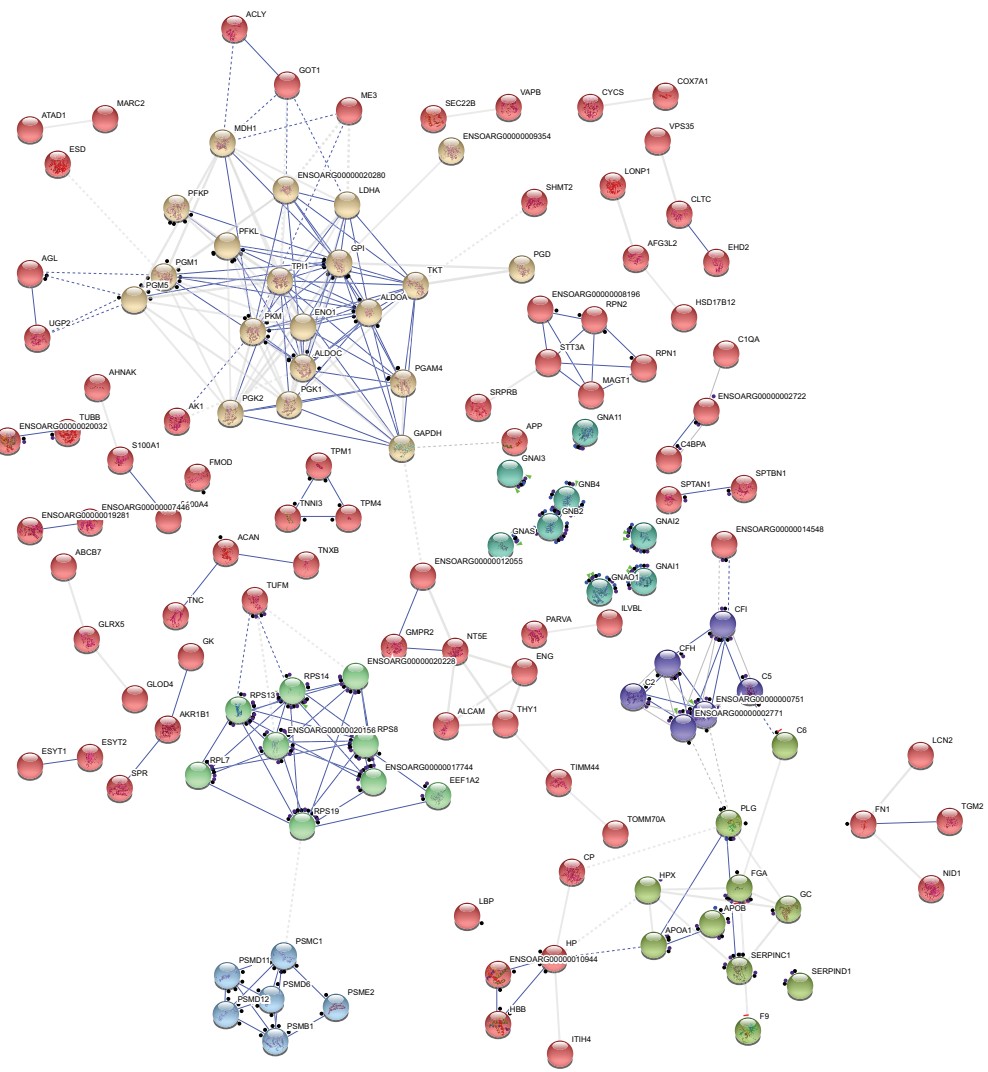

**Appendix 1—figure 2.** Interactome of 247 differentially expressed proteins showing clusters of shared action. Proteins are interconnected on the basis of their molecular action upon each other. Clusters (marked with the same colors) were generated based on protein functional similarities. The nodes represent individual proteins while molecular action is represented by blue (binding) or black (reaction). Dashed associations represent low confidence interactions, grey associations indicate an undetermined molecular action. Distance between nodes are automatically generated based on functions. Disconnected nodes were hidden for simplicity. From top left, rotating clockwise, the clusters are metabolic, G proteins, complement proteins, serpins, proteasome and ribosomal proteins.

