## [Decision Letter]

**Acceptance summary:**

The authors report remodeling changes of the tricuspid valve after experimentally-induced tricuspid regurgitation (TR) in an ovine model. They identified factors that contribute to tricuspid valve maladaptation on multiple cellular levels, including protein and matrix, and on the overall tissue scale. The work contributes to the further understanding of the pathophysiology and pathology of TR and likely will help to develop and optimize future therapeutic interventions for TR in patients.

**Decision letter after peer review:**

Thank you for submitting your article "The tricuspid valve also maladapts as shown in sheep with biventricular heart failure" for consideration by *eLife*. Your article has been reviewed by three peer reviewers, one of whom is a member of our Board of Reviewing Editors, and the evaluation has been overseen by a Senior Editor. The reviewers have opted to remain anonymous.

The reviewers have discussed the reviews with one another and the Reviewing Editor has drafted this decision to help you prepare a revised submission.

Summary:

This manuscript investigated remodeling changes in tricuspid valves after experimentally-induced tricuspid regurgitation (TR) in an ovine model. The authors demonstrate that the tricuspid valve maladapts on multiple cellular levels, including protein and matrix, and on the overall tissue scale.

The Editors consider the results presented in this manuscript novel, unique and significant. The work contributes to the further understanding of the pathology of TR and likely will help to develop or optimize therapeutic interventions for TR in the future.

There are several issues that were raised by the Editors and the reviewers that require to be addressed before this manuscript can be considered further. The most important points are summarized below.

Essential revisions:

1) It is not clear why only 'anterior leaflets' were considered, and not other leaflets as well.

2) Increase in area, height, width of leaflets. These data – like other data (thickness, collagen content) – have a lot of variation, in particular for TIC animals. For example, just by looking at the points on Figure 1B, the reviewer would say that perhaps half of the points have normal (control) ranges of width and height, and perhaps a quarter have normal (control) values of area. Are those valves perhaps the ones exposed to the dysfunctional TR the least amount of time (i.e. 13 days?). Are those valves consistently normal? (for example, are those the same valves showing control values for size, thickness and collagen)? Or are there valves that seem normal in one test (e.g. size) and are abnormal on another (e.g. thickness)? Please add to the Discussion.

3) There is data for which the individual valve variation is not given – like content of aSMA. Do these data present the same trends as other data (i.e. large variations), with TIC valves that look completely normal and others that do not?

4) Proteomics analysis found increases in metabolic and regulatory processes, which is expected if there is activation of VICs. It is not clear, though, why protein markers of EndoMT and matrix remodeling such as collagen where not listed (perhaps not found?). The authors suspect increased EndoMT. Could this be substantiated with proteomics results? How about increased collagen deposition – proteomics studies should find this as well, at least fractions of collagens. How about increases of some of the histology markers presented later (e.g. aSMA, MMP13)? These data would help support the authors' hypothesis as well as other data in the paper. Please add this data and comment on its significance.

5) What is the significance of the 'pockets' of TGFb1 in Figure 3? Are you suggesting that ECs in small vessels are undergoing EndoMT? Please clarify

6) In Figure 3A aSMA is clustered near the annulus and close to the atrialis surface. This is not really reflected in the heat map plots. Why not? Please clarify

7) It is mentioned that perhaps inflammation is responsible for some changes. Were inflammation markers found? Perhaps in the proteomics data? Please clarify

8) The authors are encouraged to provide more justification and/or discussion to the paper's interpretations of adaptation vs. maladaptation. For example, providing more discussion and/or justification for why one change would be an adaptation and another a maladaptation in terms of valve function and competence. Or alternatively in terms of surgical valve repair? And/or alternative in terms of transcatheter repair? This discussion would expand the potential readership of the article and its wider translation. In the reviewer's opinion, there are many areas for which the authors may be able to comment based on the current data and material.

9) The authors are encouraged to discuss to a greater degree the differences between findings for the TV vs. prior findings of the MV. This discussion may open up new areas of research or hypotheses. As the Authors generally lean towards in the paper, so much of knowledge in the MV has been just adopted for the TV with sometimes good and other times bad consequences. If specific areas differ in magnitude, trend, or otherwise, it would be a truly important finding for consideration. It may also provide a means to differentiate this paper as being in itself novel (which the reviewer believes) vs. just a reproduction of methods and results from the MV.

10) The animal model is great, and based on the data, reproducible. The model provides an excellent early window into FTR. TV disease on the other hand exists on a wide spectrum, with varying combinations of lesions, dysfunctions, severity that develop over years and decades. It's for the later that the Authors are recommended to consider a more cautious approach to some of the discussion relating to potential data implications and future study. Specific examples are pharmacological targeting, others on evaluating repairs, and future study recommendations.

---

## [Author Response]

Essential revisions:1) It is not clear why only 'anterior leaflets' were considered, and not other leaflets as well.

Thank you for this question. It was important to us that all analyses, such as the mechanical, histological, immunohistochemical, 2-Photon imaging, and proteomics, were performed on tissue stemming from one and the same leaflet. Of the three leaflets, only the anterior leaflet has a cusp large enough to undergo all (or most) of our extensive analyses. Additionally, because the anterior leaflet is tethered to the right ventricular free wall, the anterior leaflet is likely the most patho-mechanically stimulated leaflet thus undergoing the most maladaptation. Notwithstanding this rationale, we agree with the reviewer and look forward to exploring the relative remodeling between the leaflets in future work. We clarify our rationale now at the beginning of the Results section:

“We included only anterior tricuspid valve leaflets in our study because the anterior leaflet has the largest major cusp of the three leaflets which allowed us to conduct all (or most) of our analyses on one and the same tissue. Furthermore, because of its tethering to the remodeling right ventricular free wall we suspected that the anterior leaflet also maladapts the most of the three leaflets.”

2) Increase in area, height, width of leaflets. These data – like other data (thickness, collagen content) – have a lot of variation, in particular for TIC animals. For example, just by looking at the points on Figure 1B, the reviewer would say that perhaps half of the points have normal (control) ranges of width and height, and perhaps a quarter have normal (control) values of area. Are those valves perhaps the ones exposed to the dysfunctional TR the least amount of time (i.e. 13 days?). Are those valves consistently normal? (for example, are those the same valves showing control values for size, thickness and collagen)? Or are there valves that seem normal in one test (e.g. size) and are abnormal on another (e.g. thickness)? Please add to the Discussion.

That is a great question. We had the same question and thoughts. Therefore, we conducted two types of analyses. First, we tested correlations according to specific hypotheses about the origins of each remodeling phenomenon (thickness, stiffness, etc.), but found few statistically significant correlations (see Author response image 1). Those that were significant are already described in the Results section (e.g., Collagen content correlated positively with TR grade, Figure 4—figure supplement 1). Additionally, we took a rather crude approach in which we analyzed correlations between all measured quantities including hemodynamics measurements and geometric measurements of the ventricles. Because of the many repeated analyses, our power was naturally very low. Thus, this analysis also didn’t result in significant findings. Please see Author response image 1 for a correlative matrix that summarizes those findings. In summary, we believe that our biventricular heart failure model was an effective way to recreate the complex etiology of patients suffering from tricuspid valve regurgitation. As such, it unfortunately is also a non-transparent experimental tool in that many stimuli simultaneously acted on the leaflets creating a complex disease phenotype. In future work, we will identify other models that may be clinically less applicable but allow for isolation of stimuli, which may shed more light on the correlative/causative nature of stimuli and maladaptive phenomena. We now mention these negative findings in our limitations section of the Discussion:

“The complexity of our model is likely also reflected in our inability to find clear correlative relationships between measures of (mal)adaptation and potential (mal)adaptive stimuli, such as severity of TR or measures of right ventricular dilation.”

**Author response image 1. sa2fig1:** Experimental findings do not correlate well with clinical echocardiographic metrics of disease severity. (a-e) Correlations between echocardiography acquired metrics with experimental findings in tachycardia-induced cardiomyopathy (TIC) subjects included (a) tricuspid anterior leaflet area v. annular dilation (Pearson), (b) tricuspid regurgitation (TR) grade v. the ratio of anterior leaflet area to annular dimension (Spearman), (c) circumferential (Circ, left) and radial (Rad, right) calf stiffness v. TR grade (Spearman), (d) collagen content (Coll.) v. TR grade (Spearman), and (e) leaflet thickness v. TR grade (Spearman). Correlation coefficients (r) and p-values are inscribed in each panel, along with a linear fit (black, dashed) for visualization, if significant correlation exists. All statistical analyses here were based on specific hypotheses, and p-values thus reflect one-sided tests. (f) A correlation matrix of all combinations of TR grade, right ventricular dilation (RVD), annular dilation (AD) v. area, thickness (Thick), collagen content, calf stiffness in the circumferential (CSC) and radial (CSR) directions. Values within plot are p-values, while color map indicates the correlation coefficient (right). Correlations were Spearman’s correlation tests, unless p-value contains an asterisk (*), denoting use of a Pearson’s correlation test. Here, our significance threshold of 0.003 (Bonferroni correction) did not result in any significant correlations.

3) There is data for which the individual valve variation is not given – like content of aSMA. Do these data present the same trends as other data (i.e. large variations), with TIC valves that look completely normal and others that do not?

We appreciate the question. We saw change, at least in some regions, in all histological stains of diseased tissue. However, histological data, similarly to “other data” also had large variations in that some diseased leaflets showed considerably more positive staining than others. We clarify this now in our results of the histological results:

“Also note that, similarly to other data presented herein, immunohistochemistry stains showed heterogeneity among TIC subjects. That is, some TIC leaflets showed more positive staining than others.”

4) Proteomics analysis found increases in metabolic and regulatory processes, which is expected if there is activation of VICs. It is not clear, though, why protein markers of EndoMT and matrix remodeling such as collagen where not listed (perhaps not found?). The authors suspect increased EndoMT. Could this be substantiated with proteomics results? How about increased collagen deposition – proteomics studies should find this as well, at least fractions of collagens. How about increases of some of the histology markers presented later (e.g. aSMA, MMP13)? These data would help support the authors' hypothesis as well as other data in the paper. Please add this data and comment on its significance.

Again, we appreciate the reviewer’s suggestion. In total, we found 247 genes/proteins whose expression changed statistically significant. We limited our reporting to those that created clusters, but referred the readers to a table that listed every protein. We recognize that including not only the proteins that formed clusters, but also individual ones that support our other data would have been helpful. Therefore, we are now also specifically mentioning the upregulation of genes related to matrix remodeling, such as COL6A3, which is related to the alpha-chain of a type VI collagen. Similarly, we also specifically refer to FN1, GPC4 and PGLYRP1, all proteins involved in synthesis and regulation of glycans. Similarly, we are now also reporting on the upregulation of specific proteins related to the mesenchymal phenotype, e.g., CD14, CD163, FERMT2, PLS3, and S100A1, see also response to 7. However, no proteins related to endothelial marker expression were identified. The potential reasons for this absence of findings are multi-fold, but most likely related to either lack of statistical power or mismatch in temporal occurrence. In other words, it is possible that Endo-MT had occurred before our observation time point and related soluble proteins where therefore not present anymore. We added a Discussion point to mention this potential limitation:

“Specifically, we did not find CD31+ staining cells co-localized with αSMA, as reported by Bartko et al., (Bartko et al., 2017). Similarly, we found no significant upregulation of proteins related to EndoMT in our proteomic analysis. While the reasons could be multi-fold, one possible explanation could be that EndoMT may have occurred before or after our observational period.”

5) What is the significance of the 'pockets' of TGFb1 in Figure 3? Are you suggesting that ECs in small vessels are undergoing EndoMT? Please clarify

Thank you for your question and great observation. We apologize for this confusion in Figure 3. We include this observation for two reasons. First, previous studies on the mitral valve have shown that leaflet remodeling was accompanied by neovascularization. Thus, we wanted to highlight similar observations in the tricuspid valve. We believe the TGFb1 positive staining is likely related to the (possible) angiogenic signaling and therefore only indirectly a result of the tissue maladaptation. We recognize that we failed to include these remarks, and now include them in the figure legend: “(d) interstitial collection of TGF-ꞵ1 in ‘pocket’ which may be a neo-microvessel. Note, positive TGF-ꞵ1 staining in this pocket could be related to angiogenic signaling.

6) In Figure 3A aSMA is clustered near the annulus and close to the atrialis surface. This is not really reflected in the heat map plots. Why not? Please clarify.

The heat map indicates an almost three-fold increase in aSMA near the atrialis and near the annulus (top left rectangle). It is possible that our labelling may have caused confusion. We are NOT showing the entire leaflets in the histology images, from annulus to free edge. Instead, we are showing segments of the leaflets, which we believed best represented our findings within the heat maps. In other words, the histology images in Figure 3A show only the leaflet segment near the annulus. With this in mind, the reviewer may see that our histology images and our heat map are well aligned? We recognize that our labelling is not ideal and we added subpanels in Figure 3 to clearly indicate which part of the leaflet we are showing.

7) It is mentioned that perhaps inflammation is responsible for some changes. Were inflammation markers found? Perhaps in the proteomics data? Please clarify

Thank you for this question. Classic inflammatory cytokines involved in heart valve disease, such as TGFb, were not found by proteomics as they are typically present in small cellular concentrations. However, our study did find coagulation factor F9 highly up-regulated, along with a number of complement proteins (C2, C5, C6 and others), indicating the increased production of pro-inflammatory mediators downstream. We edited our text to include these findings explicitly:

“Compared to CTL, TIC anterior leaflets overexpressed metabolic proteins (ENO1, ESD, ALDOA, ALDOC, ASPH, UGP2, AK1, PGK1, PGK2, PKM, LDHA, PGD, among others), serpins (A1, A5, C1, D1, and others), apolipoproteins (A1 and B), proteins related to matrix remodeling, such as COL6A3, which is tied to the alpha-chain of a type VI collagen, and FN1, GPC4 and PGLYRP1, all proteins involved in synthesis and regulation of glycans. Additionally, we found upregulation in specific proteins related to the mesenchymal phenotype, e.g., CD14, CD163, FERMT2, PLS3, and S100A1. Finally, we found that coagulation factor F9 was highly upregulated, along with a number of complement proteins (C2, C5, C6 and others), indicating the increased production of pro-inflammatory mediators downstream.”

8) The authors are encouraged to provide more justification and/or discussion to the paper's interpretations of adaptation vs. maladaptation. For example, providing more discussion and/or justification for why one change would be an adaptation and another a maladaptation in terms of valve function and competence. Or alternatively in terms of surgical valve repair? And/or alternative in terms of transcatheter repair? This discussion would expand the potential readership of the article and its wider translation. In the reviewer's opinion, there are many areas for which the authors may be able to comment based on the current data and material.

We much appreciate the suggestion and recognize that we have not been as clear and detailed as we could have been. We now expand our Discussion to include a perspective on what we believe the differences are and how/why there are relevant to a broader readership:

“If true, our current and future work, should focus on delineating the potentially differing mechanistic origins of tricuspid valve adaptation and maladaptation. […] Such strategies could have broad impact by enabling optimization of both surgical repair techniques and interventional approaches, such as transcatheter annuloplasty (Kuwata et al., 2017) or leaflet clipping (Fam et al., 2018).”

9) The authors are encouraged to discuss to a greater degree the differences between findings for the TV vs. prior findings of the MV. This discussion may open up new areas of research or hypotheses. As the Authors generally lean towards in the paper, so much of knowledge in the MV has been just adopted for the TV with sometimes good and other times bad consequences. If specific areas differ in magnitude, trend, or otherwise, it would be a truly important finding for consideration. It may also provide a means to differentiate this paper as being in itself novel (which the reviewer believes) vs. just a reproduction of methods and results from the MV.

This is an excellent point. Of course the reviewer is correct in that we have taken inspiration from our own work on the mitral valve as well as that by others (especially Levine and colleagues, whom we also cite). We have now added a separate Discussion point solely designated to contrasting our findings to those in the mitral valve:

“We set out to identify whether the tricuspid valve, like the mitral valve, (mal)adapts. […] While the reasons could be multi-fold, one possible explanation could be that EndoMT may have occurred before or after our observational period.”

10) The animal model is great, and based on the data, reproducible. The model provides an excellent early window into FTR. TV disease on the other hand exists on a wide spectrum, with varying combinations of lesions, dysfunctions, severity that develop over years and decades. It's for the later that the Authors are recommended to consider a more cautious approach to some of the discussion relating to potential data implications and future study. Specific examples are pharmacological targeting, others on evaluating repairs, and future study recommendations.

We very much appreciate the reviewer’s words of caution. We reviewed our Discussion section and agree. We edited our limitation sections and others to be more cautious. For example:

“As we fail to capture the late response of the tricuspid valve to disease mechanisms, we ask the reader to extrapolate our findings with caution.” Or “Although, direct comparison should be done with caution as our current model differs from those of mitral valve (mal)adaptation.” Or “While much remains to be understood, we hope that we have taken a first step toward similar studies that may support pharmacological treatment of TR surgery and/or intervention.”